# Insight into One Health Approach: Endoparasite Infections in Captive Wildlife in Bangladesh

**DOI:** 10.3390/pathogens10020250

**Published:** 2021-02-23

**Authors:** Tilak Chandra Nath, Keeseon S. Eom, Seongjun Choe, Shahadat Hm, Saiful Islam, Barakaeli Abdieli Ndosi, Yeseul Kang, Mohammed Mebarek Bia, Sunmin Kim, Chatanun Eamudomkarn, Hyeong-Kyu Jeon, Hansol Park, Dongmin Lee

**Affiliations:** 1Department of Parasitology, School of Medicine, Chungbuk National University, Cheongju 28644, Korea; tilak.parasitology@sau.ac.bd (T.C.N.); keeseon.eom@gmail.com (K.S.E.); vetazmo@gmail.com (S.C.); barakan@gmail.com (B.A.N.); jeonhk@cbnu.ac.kr (H.-K.J.); 2Department of Parasitology, Sylhet Agricultural University, Sylhet 3100, Bangladesh; saiful1201082@gmail.com; 3International Parasite Resource Bank, Cheongju 28644, Korea; 4Rangpur Zoological and Recreational Garden, Rangpur 5404, Bangladesh; dr.hmshahadat@gmail.com; 5Parasite Research Center, Chungbuk National University, Cheongju 28644, Korea; paraseul47@gmail.com (Y.K.); bia_vet39@yahoo.fr (M.M.B.); claudine@chungbuk.ac.kr (S.K.); chatanune@yahoo.com (C.E.)

**Keywords:** endoparasites, captive wildlife, one health, *Spirometra decipiens*, Bangladesh

## Abstract

*Introduction*: Endoparasites in captive wildlife might pose a threat to public health; however, very few studies have been conducted on this issue, and much remains to be learned, especially in limited-resource settings. This study aimed to investigate endoparasites of captive wildlife in Bangladesh. Perception and understanding of veterinarians regarding one health and zoonoses were also assessed. *Materials and Methods*: A cross-sectional study was conducted from October 2019 to August 2020. A total of 45 fecal samples from 18 different species of wild animals (i.e., 11 species of mammals: herbivores, carnivores, and omnivores, six birds, and a single reptile species) were collected randomly. Parasitological assessments were done by modified formalin ether sedimentation technique and rechecked by Sheather’s sugar floatation technique. Molecular identification of *Spirometra* spp. was conducted by amplifying the cytochrome *c* oxidase 1 (*cox*1) gene. Questionnaire surveys among 15 veterinarians and an in-depth interview (IDI) with a zoo officer were conducted. *Results*: Helminths (*Spirometra* sp., *Capillaria* sp., *Ascaridia*/*Heterakis*, opisthorchiid, strongyles, acuariid, hookworms, roundworms, and unidentified nematode larvae) and protozoa (coccidian oocyst) were identified, and the overall prevalence was 48.9% (22/45). The *cox*1 sequences (341 bp) of the Bangladesh-origin *Spirometra* species from lion showed 99.3–99.7% similarity to the reference sequences of *Spirometra decipiens* (GenBank No: KJ599679.1; MT122766). The majority of study participants (86.6%) agreed about the importance of endoparasite control in zoo animals, and 73.3% expressed that the one health concept should be promoted in Bangladesh. Only 6.7% of veterinarians perceived confidence in diagnosing parasitic diseases and preventing antiparasiticidal resistance. *Conclusions*: In the present survey, we found a considerable prevalence of endoparasites in captive wildlife. For the first time, zoonotically important *S. decipiens* from lion was molecularly characterized in Bangladesh. Veterinarian training is required to improve parasite control knowledge and practice. This study highlights the need for routine parasitological assessment, promotion of one health, and improvement of the implementation of current parasite control strategies in zoo animals.

## 1. Introduction

One health is a collaborative, multisectoral, and transdisciplinary approach to promoting, improving, and protecting the health and wellbeing of all species by strengthening communication and collaboration between health professionals [1,2]. The World Organization for Animal Health (OIE) incorporates this concept as a collaborative global approach to understanding risks to humans, animals, and ecosystem health. Wildlife health, as defined by Stephen [3], is a dynamic social construct based on human expectations and knowledge. Health and sustainable maintenance of wildlife are mutually interdependent with the surrounding community and environment. The one health movement has heightened this focus by recognizing wildlife as a major source of zoonotic infections, thereby encouraging intensive research in the detection of wildlife pathogens [1,3]. As the natural habitats of many wild animals have been altered, degraded, or occupied by the growing human population, between 1970 and 2016, more than two-thirds of the global wildlife population has declined [4]. Zoological gardens or zoos exist to protect endangered animal species and to evaluate the need for biodiversity protection. Zoos also play a crucial role for aesthetic, recreational, and educational purposes [5,6]. However, wild animals maintained in captivity present challenges; controlling infectious diseases and maintaining good husbandry are essential [5]. Animals kept in crowded cages and in limited environments are stressed; the immune systems of animals become weak, making animals more susceptible to parasitic infections [6,7]. Animals with a regular deworming program usually do not show any major sign or symptoms of parasitic infection [6,7]; however, some endoparasites of wild animals may threaten the health of zoo workers, visitors, and surrounding city dwellers. Knowledge of the disease status of the zoo animals and appropriate screening are essential for public health safety and for the wellbeing of animals, as many pathogens can infect multiple host species [5,8].

The successful implementation of any disease control program depends on adequate and sensitive methods for detecting and monitoring infections. Epidemiological data are also critical; however, application of such data is not easy and has to be done with care [9]. Understanding the relationships among biodiversity, health, and disease enhances policy development opportunities and ensures biodiversity conservation [10]. Veterinarians in zoos are leading to deal with health and management issues that threaten the long-term survival of wildlife species. Veterinarians come in direct contact with wild animals during clinical examination, specimen collection, biopsy procedure, surgery, hand-rearing of newborn animals, and postmortem examinations. A clear understanding of veterinarians about the interconnection between human, animal, and environmental health is needed to prevent and control zoonotic diseases [11]. To ensure public health safety and the protection of biodiversity, the combined efforts of human, animal, and ecosystem health is necessary. This combined effort is termed the “one health approach” and has become widely accepted within the health professions [12].

Almost all zoos in Bangladesh are located near city areas due to economic reasons, and close contact exists between human and captive animals. Animals are housed in close proximity to each other due to limited space and may serve as sentinels of disease for humans, wildlife, and domestic animals [13]. This could also be a threat to public health due to high household density around the areas where substantial barriers between zoo and locality are absent. So far, very few studies have been conducted on the parasites in captive wild animals in Bangladesh, and very little information is known about the occurrence of endoparasites. This cross-sectional pilot study was conducted as a part of the one health approach to assess endoparasites in captive animals in Bangladesh. The perception and understanding of veterinarians regarding one health were also assessed. Findings from this study may help zoo authorities to devise control strategies and raise awareness in related communities.

## 2. Results

Commonly given foods and antiparasitic medications to examined animals are shown in Table 1. A variety of foods are given to the animals according to their nutritional requirements depending upon the species, age, and physiology. In addition to grasses and plants, foodstuffs given to animals are pelleted formulated feeds, breads, fruits, vegetables, meat, and dairy products. A routine deworming program with several types of antiparasitic drugs is performed every 3–4 months to manage parasitic infections.

Fecal examination results are shown in Table 2 and Figure 1. We found a high infection rate; 22 out of 45 samples (48.9%; 95% confidence interval = 36.1–59.0%) were positive for parasitic infections. Out of 18 examined host species, 12 species were found positive for endoparasites. Among 45 examined animals, 12 animals showed mono-parasitic infection, while 10 animals were co-infected with different helminths and protozoa species. Concerning helminths, we identified gastrointestinal strongyles eggs in donkey, monkey, and hanuman (7/19, 36.8% of herbivores), *Toxocara* sp. in lion (2/10, 20.0% of carnivores), *Spirometra* sp. in lion and African civet (3/10 30.0% of carnivores), opisthorchiid eggs in tiger (1/10, 10.0% of carnivores), *Baylisascaris transfuga* in black bear (1/10, 10.0% of carnivores), *Capillaria* sp. in turkey and peacock (4/15, 26.7% of avian species), hookworm eggs in African civet and black bear (3/10, 30.0% of carnivores), roundworm eggs in vulture and peacock (3/15, 20.0% of avian species), and acuariid eggs in peacock (1/15, 6.67% of avian species). Furthermore, we found roundworm eggs in vulture and unidentified nematode larvae in emu. In case of protozoa, we found an oocyst of *Eimeria* spp. in spotted deer, turkey, and peacock (4/15, 26.7% of avian species; 1/29, 3.5% of mammals) and an unidentified coccidian oocyst in emu (1/15, 6.7% of avian species).

During laboratory examination, ovoid-shaped eggs sized about 61–65 × 31–33 μm with pointed ends were observed in the fecal samples of lion. According to these morphological characteristics, the parasite was assigned to *Spirometra* sp. We extracted DNA from *Spirometra* eggs, and sequence analysis based on the mitochondrial *cox*1 gene (341 bp) showed 99.3–99.7% similarity to the reference sequences of *S. decipiens* (KJ599679.1; MT122766) and 90% similarity to *S. erinaceieuropaei* (KJ599680). A phylogenic tree constructed using maximum likelihood (ML) methods also showed the *Spirometra* sp. egg recovered in the present study clustered with *S. decipiens* (Figure 2).

An in-depth interview (IDI) with a zoo officer revealed that the leading barriers related to disease management were lack of training in zoonotic diseases and laboratory practices, lack of policy for workplace safety, absence of promotion activities on one health, and insufficient research on parasitic diseases. The following was mentioned:

“I am aware about my contributions to protect these valuable animals, as well as to ensure public health safety, but I did not get enough chance to attend training and information is also deficient… We routinely administer anthelmintics to all animals, but laboratory facilities for fecal examination or efficacy monitoring are yet to be developed… laboratory personnel are also lacking at this moment.”

Of the veterinarians surveyed (Appendix A), the majority agreed/strongly agreed (80.0%) parasites could be transmitted to humans/animals from the environment and vice versa; however, only 6.7% had enough confidence to diagnose, identify, and differentiate parasitic diseases. Regarding one health, the majority (86.6%) agreed that one health could effectively control parasitic infections, and 73.3% expressed that the one health concept should be promoted by the national and local government in Bangladesh. Unfortunately, most of the respondents stated that resources to learn zoonoses and one health are lacking. Furthermore, most respondents perceived that they did not have had a sufficient understanding of antiparasiticidal resistance. 

## 3. Discussion 

The control of parasites is an essential feature of zoo management, as helminth infection can adversely impact the health of captive animals. Zoo animals are parasitized by many endoparasite species [7,14,15,16] and, in this study, several parasite species were identified in mammals and birds. Most endoparasites usually cause minimal discomfort to healthy animals; however, in conditions such as high population density, stress, and adaptation to new environments or extended periods in a confined space, these infections can cause disease [14]. Nearly half of the examined samples were positive for at least one parasitic species in this study. In zoos, control of these parasites should primarily focus on lessening infection pressures by administering anthelmintics. Although animals were treated with antiparasitic drugs periodically, the considerable infection rate observed in this study indicates that the current preventive measures applied are insufficient or that a high percentage of reinfections occur. Lack of quarantine, when new entered animals and existing animals are put together, and free-ranging stray animals could also be possible source for parasitic infection. Moreover, the frequent use of broad-spectrum anthelmintics combined with inappropriate dose and administration methods may lead to the development of resistance to anthelmintics [17]. In an experiment conducted by Young et al. [18], several gastrointestinal strongyles in captive wild ruminants showed a higher level of resistance to common anthelmintics such as levamisole and avermectin. To ensure anthelmintic efficacy, revision of parasite control strategies is required, including a greater emphasis on surveillance through periodic parasitological assessment.

The eggs/oocysts of 12 different types of parasites were observed in the study, and almost half of the examined animals were found infected with at least one parasite species, with some of them harboring potentially zoonotic parasites. Most of the parasites found in this study have fecal–oral routes of transmission that have commonly been reported in captive animals in various zoological gardens by several authors [7,15,19]. In most cases, animals may acquire the infections through contaminated foods, water, and even staff and visitors of the zoo. Humans have been reported to play a role in transmitting parasites through their shoes, clothes, hands, food, or working tools and act as vectors [20]. The environment can also play an important role in the transmission of parasites and can be contaminated with parasitic eggs or larvae through improperly disposed fecal materials [20]. Feeding raw or undercooked fish and meat might be a potential source for trematode and cestode infection [21] as tissues of intermediate or paratenic hosts contain larval forms of parasites. In this study, interviews with a zoo officer revealed that animal cages and facilities were cleaned and fecal materials were removed from animal cases regularly. However, during observation, indoor and outdoor facilities were not found to be cleaned adequately. Preventing environmental contamination by eggs and larvae of parasites could be one of the key steps to stopping the transmission of parasites to the wildlife. Regular monitoring for sanitation and cleaning is necessary to minimize the risk. Zoos also need to take measures to exclude rodents such as rats and mice from food storage and preparation areas, as those rodents can serve as a reservoir host of many zoonotic helminths such as *Baylisascaris* spp., *Toxocara* sp., hookworms, and *Capillaria hepatica* [22,23].

The present findings show that parasites can be common in zoo birds since four out of six examined avian species were coprologically positive and most of the infected birds had mixed infections. Papini et al. [20] mentioned that zoo birds were about 15 times more likely to develop mixed infections than their pet counterparts. In zoos, closely related species were all housed together, and many bird nematodes can have a wide host range. We observed *Ascaridia* spp./*Heterakis* spp. eggs in the fecal samples of avian species. However, these two genera of nematodes cannot be differentiated on the basis of egg morphology and size [24]. Ascarids are included in the most important intestinal nematodes infecting birds, since they are often the cause of intestinal obstruction, intestinal perforation, and death of infected birds [25]. In most birds, coccidiosis may be caused by species of the genera *Eimeria* or *Isospora*, and species of both genera have been reported in emu [26,27,28]. The differentiation of these two genera is based on the morphology of mature (sporulated) oocysts, since mature *Eimeria* oocysts show four sporocysts each containing two sporozoites, while mature *Isospora* oocysts show two sporocysts each containing four sporozoites [29]. Therefore, further research is required to identify the species infecting the emu in this study. We were not able to identify the nematode species found in emu feces which measured 2.1–2.4 mm in length by 0.79–0.84 mm in width. However, according to its morphology, this nematode could be either a free-living stage of *Dromaestrogylus* sp. or a free-living nematode species.

Helminth infection was more common than protozoal infection, with helminth eggs observed in 21 (46.7%) animals, while protozoans were observed in four (8.9%) of the total positive animals. Interestingly, only one cestode and one trematode were detected in this study. The probability of trematode and cestode infections in captive animals is lower, as the life cycle of these helminths is indirect; they require one or more intermediate hosts for their development and transmission [30,31]. Since animals in zoos are kept in closed enclosures, giving very limited access to the intermediate hosts, their intermediate hosts are less likely accumulated in the enclosure [32]. However, in our study area, strong barriers between the outer and inner environment of the zoo were absent, and this could be the reason for the transmission these parasites. Feeding tissues of larva-infested intermediate or paratenic hosts could be the source of infection in these cases; animals could have been infected with cestode or trematode larvae through the consumption of raw or undercooked fish or meat, or they could have become intermediate hosts when accidentally ingesting worm eggs [21,33]. Most of the parasites of wild animals, especially from carnivores, can also infect other animals and humans; therefore, they have public health importance [9]. Helminths found in this study, such as *Baylisascaris* sp., *Spirometra* sp., *Toxocara* sp., hookworms, and opisthorchiid, are well-recognized zoonotic agents [6,15,21,34]. Although not zoonotic, other helminth species identified in this study, especially from wild herbivores, may also infect other mammal hosts [18]. Therefore, it is emphasized that visitors, veterinarians, and zoo staff who have contact with these animals must take precautions to avoid infection and possible zoonotic transmission.

Wild animals play a critical role in the epidemiology of helminths, as they may be considered as a potent reservoir for many roundworms with implications in constant transmission to human populations and pet animals [34]. One of the most important zoonotic helminths observed in bears is *Baylisascaris* sp. *Baylisascaris* species infects more than 50 different animal species, including humans and wild mammals. Although egg morphology alone is not sufficient to specify *B. transfuga*, other ascarid nematodes can be excluded from the differential diagnosis on the basis of epidemiology and available studies. *B. transfuga* is a ubiquitous roundworm, reported worldwide, which exclusively infects bears both free ranging and in captivity [35,36]. Eggs of this parasite have a thick, impermeable, desiccation-resistant lipid layer [37] and can remain infective for at least 15 months under artificial conditions [38]. Humans, especially zoo staff and trappers, can be infected from a contaminated environment. In humans, *Baylisascaris* larvae may cause ocular and visceral *larva migrans*, which may become fatal when larvae invade the central nervous system [22]. Other important potentially zoonotic parasites identified in this study were *Toxocara* spp. infecting the lion, i.e., *T. cati* which is transmitted when infective larva-infested eggs are ingested by human hosts [39]. The disease is distributed worldwide, but particularly highly in tropical lower-resource countries. It causes significant morbidity and poses an important, yet largely unaddressed public health problem in areas of high prevalence [40]. The majority of *Toxocara* infections remain asymptomatic, whilst a minority develop fatal diseases of visceral *larva migrans*, ocular *larva migrans*, and covert toxocariasis. The eggs hatch in the small intestine, and the larvae migrate to the liver, lungs, eyes, and other body organs, where they cause tissue necrosis, chronic liver disease, edema, hemorrhage, and eosinophilia [9,39]. Hookworms can pose serious public health hazards and display a high pathogenic potential due to their blood-feeding behavior [34]. Human infection with hookworm species commonly causes cutaneous *larval migrans*, with painful, itchy eruptions along the path of migrating larvae [9,16]. In the lungs, these larvae infiltrate the alveoli and migrate up to trachea, from where they reach the intestine. People involved in direct contact with soil and animals are at risk of acquiring hookworm infection [34,41]. Therefore, it is important that zoo staff adopt proper hygiene, including hand washing and wearing personal protective equipment, to avoid contact with hookworm larvae. In this study, zoonotically significant opisthorchiid eggs were also identified from tiger. Opisthorchiid flukes, especially *Clonorchis* sp. and *Opisthorchis* sp., infect millions of people, particularly in Southeast Asia, Eurasia, and North America [42]. We did not distinguish the genera of opisthorchiid flukes as egg morphometry alone fails to accurately distinguish the species [42]. Both genera were reported from *Panthera* spp. Seryodkin et al. [43] reported *C. sinensis* from Siberian tigers (*Panthera tigris altaica*) and Stuti et al. (2012) [44] reported *Opisthorchis* sp. from leopard (*Panthera pardus*). However, no study was found regarding opisthorchiid infection in Royal Bengal tiger (*Panthera tigris*). Eating raw or undercooked fish, coupled with the inadequate sanitation facilities, provides a perfect environment for the spread of opisthorchiid infections. Improper disposal of faces leads to the continuation of the life cycle by contamination of the environment and infection of intermediate hosts [9,43]. *Spirometra* spp. represent another neglected parasitic agent with zoonotic potential, since they are the cause of sparganosis in humans and animals. Sparganosis has been reported worldwide, but most cases occur in Asia and Americas [33,45,46,47]. Human infection may occur in various ways, including drinking contaminated water containing larvae or first intermediate hosts and ingesting raw or undercooked meat from infected intermediate secondary hosts or paratenic hosts (frogs, snakes, and game such as feral swine) [45]. Almost all zoos in Bangladesh are located in urban and crowded areas with inadequate sanitation facilities that favor the transmission of zoonotic parasites. Moreover, loose animals such as rodents, birds, and even stray dogs and cats may also contribute to the transmission of parasites to animals kept in the zoo. 

Although human sparganosis has been reported in neighboring countries such as Myanmar [48] and India [49], *Spirometra* information in Bangladesh is limited, and genetic analysis has not been done on the species. From a heavily infected lion, we successfully extracted the DNA of the spirometriid eggs isolated from fecal samples. According to the mitochondrial *cox*1 sequences available in the NCBI (National Center for Biotechnology Information) database, this study has shown that the Bangladesh-originated *Spirometra* species can be identified as *S. decipiens.* However, the taxonomy of the genus *Spirometra* is still unclear, and there is considerable controversy in the literature [47]. Although *S. decipiens* from Korea [50], *S. ranarum* from Myanmar [48], and *S. erinaceieuropaei* from China [45] were reported and claimed as individual species, some researchers reported *S. decipiens* and *S. ranarum* as synonyms of *S. erinaceieuropaei* [51,52]. In addition to morphological differences, Jeon et al. [48] analyzed the mitochondrial genomes of three *Spirometra* spp. and showed nucleotide divergence among *S. decipiens, S. erinaceieuropaei*, and *S. ranarum*. Recently, Yamasaki et al. [47] re-examined all three *Spirometra* species on the basis of the mitochondrial *cox*1 gene and stated that *S. decipiens* and *S. ranarum* from Asia are probably conspecific with *S. mansoni*. In this regard, further studies are needed to clarify the taxonomic status of *Spirometra.*

One health offers a valuable platform for implementing policies and strategies in the health biodiversity context to address the public health challenges of the 21st century [2,12]. Zoonotic diseases are increasingly recognized as a serious threat to public health, and it is estimated that 70% of zoonotic diseases originate from wildlife. An estimated 700 million people worldwide visit zoos each year [53], and interaction between humans and captive animals is increasing gradually with the potential of zoonotic disease transmission. Understanding disease transmission as a whole and understanding the role of the captive animals in human, animal, and environmental health is considered a vital requirement for the one health approach. However, to design and implement control measures, it is necessary to identify and understand the opportunities and challenges that exist in the contextual settings in Bangladesh. Veterinarians are often placed in a fundamental position to maintain relationships of captive animals with humans and other animals, as well as their effect on the surrounding environment, disease spread, natural resource availability, culture, and society. Appropriate knowledge of veterinarians regarding zoonosis and one health will ensure animal welfare and broaden the perspective to maintain all activities in a healthy manner [54]. In this study, we sought to determine the attitude, perception, and understanding of veterinarians concerning parasitic zoonoses and one health. Respondents showed a good perception of parasitic zoonoses and a positive attitude that one health approach can effectively ensure animal and human health. However, practical frameworks for professionals to prevent parasitic zoonosis and promote the one health approach are lacking in Bangladesh. Zoo workers do not have sufficient understanding and equipment to protect themselves from zoonoses at the workplace. Despite a good attitude, there is a need to improve their knowledge to enhance public health security. For effective prevention and control of zoonotic diseases, veterinarians need to understand the interactions among humans, animals, and the environment [11].

There are several limitations as our study aimed to create a baseline inventory of key endoparasites species infecting captive animals in Bangladesh. It was not our intention to provide a comprehensive overview of all species or to provide species-level identification. Due to a deficiency of samples, we were unable to extract DNA from all fecal samples. Furthermore, we were unable to pinpoint the source of infection in most animals; in most cases, zoo authorities had no data regarding the origin of the animals. Despite several limitations, we identified several parasite species that will increase attention in future research.

## 4. Materials and Methods

### 4.1. Study Area, Study Population, and Sample Collection

The study was conducted from October 2019 to August 2020. Forty-five fresh fecal samples were collected from 18 different species of mammals, birds, and reptiles captured at a public zoo in Bangladesh. Samples were collected from clinically healthy animals only (Table 1). Symptomatic and isolated animals were excluded as per regulation of the zoo administration. All samples were fixed with 10% formalin and 70% ethanol. Fixed samples were transported to the Department of Parasitology, School of Medicine, Chungbuk National University, South Korea. The questionnaire survey was conducted among 15 veterinarians, and an in-depth interview was conducted with a zoo officer.

### 4.2. Coprological Examination

Formalin-fixed fecal samples were processed for examination. For the modified formalin ether sedimentation technique, the procedure began with thorough mixing of the fecal suspension and sieving through gauze into a test tube. The test tube was then vigorously shaken, followed by the addition of 3 mL of ether as an extractor of fat and fecal debris. After centrifugation for 5 min at 1500 rpm, the supernatant was discarded, and the sediment was examined using a light microscope. For Sheather’s sugar floatation technique, the fecal sample was suspended in distilled water and sieved with a double layer of gauzes, transferred to a test tube, and centrifuged at 1500 rpm for 5 min. The supernatant was then poured off, and Sheather’s sugar solution (approximately 1.27 specific gravity) was added as a flotation solution. The mixture was then vigorously shaken and centrifuged at 1000 rpm for another 10 min. The tube was filled with Sheather’s sugar solution up to the upper meniscus and covered with a coverslip. About 15 min later, the coverslip was removed, and the sample was placed onto a glass slide and examined under the microscope. The identification of eggs, ova, and larva was made on the basis of standard keys such as size, shape, nature of the shell, and nature of germinal cells [24,55]. 

### 4.3. PCR Amplification and Sequencing for Zoonotically Significance Eggs

DNA extraction was done from the lion’s fecal sample (fixed with 70% ethanol) that was microscopically positive for *Spirometra* sp. eggs, using the QIAamp DNA Stool Kit (Qiagen, Hilden, Germany), according to the manufacturer’s instructions. However, elution was repeated twice using distilled water instead of elution buffer. The concentration and purity of DNA were measured (NanoDrop Spectrophotometer, Thermo Fisher Scientific Solutions Co. Ltd., Seoul, Korea), and it was stored at −20 °C before use. A region within mitochondrial cytochrome *c* oxidase subunit I (*cox*1) was amplified and sequenced. PCR and DNA sequencing were performed according to established protocols. The PCR primers used were p1f (5′–TGGTTTTTTGGACATCCTGAA–3′) and p1r (5′–ATCACATAATGAAAGTGAGCC–3′) [56]. The PCR reactions were performed in a Kyratec PCR Thermal Cycler (Queensland, Australia). The PCR was carried out in a final reaction mixture containing 30 μL, including 1 μL of each primer (10 pmol), 1 μL of generic DNA, 8 μL of 5× PCR Master Mix (ELPIS biotech, South Korea), and 19 μL of distilled water. A negative control was applied in each run. PCR conditions were as follows: 94 °C for 3 min; 94 °C for 1 min, 48 °C for 1 min, and 72 °C for 1 min for 35 cycles; 72 °C for 10 min. When amplifications did not work adequately, the annealing temperature was changed and adjusted. The PCR products were run on a 1% agarose gel and visualized using an ultraviolet (UV) transilluminator. DNA sequencing was performed by a company (Cosmogenetech, Seoul, Korea). The obtained sequences were assembled with Geneious program 9.0 (Biometer, Auckland, New Zealand). Sequences were aligned using ClustalW multiple alignments implanted in MEGA7 [57,58]. Alignments were trimmed to the length of the shortest sequence. Sequencing analysis was carried out by BLAST (Basic Local Alignment Search Tool) algorithms and databases from the National Center for Biotechnology Information. Phylogenetic trees were constructed with four taxa of Diphyllobothriidae comprising *S. decipiens* (GenBank no. KJ599679; MT122766; LC328899), *S. erinaceieuropaei* (GenBank no. KJ599680), and *Dibothricephalus latus* (=*Diphyllobothrium latum*, GenBank no. NC008945; as outgroup), using maximum likelihood (ML) algorithms. Bootstrap values were calculated using 1000 replicates. The multiple alignments were performed with the program Muscle [59] and Hasegawa–Kishino–Yano (HKY + G) was chosen according to the Modeltest using MEGA7 [57]. To describe the best substitution patterns, the lowest BIC scores (Bayesian information criterion) were considered. The generic DNA used in the present study was stored in the International Parasite Resource Bank, Korea (PRB001200).

### 4.4. Data Collection and Analysis

A semi-structured questionnaire survey among 15 veterinarians was conducted to access knowledge and perception of zoonoses and one health. One in-depth interview with a zoo officer was also conducted to understand the current management and practices required to facilitate sustainable parasite control regimens. Interviews were conducted through an online platform due to government regulations of social distancing to prevent COVID-19. The semi-structured questionnaire consisted of 15 questions (four open-ended; 11 close-ended), and it was designed to obtain information related to study objectives. Respondents were given the opportunity to include additional information for some open-ended questions. An online questionnaire was created using web-based software, Survey Monkey, and the link was distributed via email to respondents. The IDI was conducted using an interview guide designed to assess the level of knowledge, attitude, practices, awareness, current strategies, existing barriers, opportunities, challenges, and future suggestions associated with parasite control and implementing the one health approach. The IDI was conducted through an online platform (zoom.us) and lasted about 40 min. All collected data (both questionnaire and transcribed text) were double-checked by research assistants and rechecked by the principal researcher for missing data or incorrect entries. Frequency tables and cross-tabulations were produced for each study variable. Data were analyzed using STATA 13.1 data analysis software [60]. Descriptive statistics were used to tabulate and describe the data.

### 4.5. Ethics Permission

The study protocol was reviewed and approved (Approval No. EC/2020/59) by the Department of Parasitology, Chungbuk National University, Korea, and the Department of Parasitology, Sylhet Agricultural University, Bangladesh. Registered veterinarians collected the study materials.

## 5. Conclusions

In this study, a considerable prevalence of endoparasites was observed in captive wildlife in Bangladesh. Some parasites found in this study are well known as human pathogens and might be a potential source of zoonotic transmission. Participants showed a good perception of zoonoses and one health; yet, relative education and capacity building training are required to translate this perception into practice. To our knowledge, this is the first molecular identification of *S. decipiens* in Bangladesh, and further study to determine the exact taxonomic status of *Spirometra* species in this area is required. Results from this study emphasize the need for integrated approaches with greater emphasis on regular parasite surveillance and monitoring to ensure sustainable parasite control.

## Figures and Tables

**Figure 1 pathogens-10-00250-f001:**
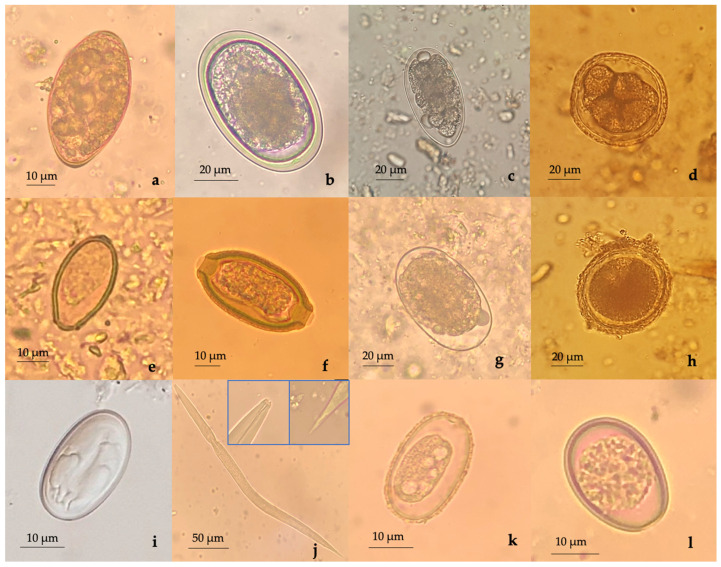
Identified endoparasites in captive wildlife: (**a**) *Spirometra* sp. egg (lion); (**b**) roundworm egg (peacock); (**c**) intestinal strongyle egg (monkey); (**d**) *Baylisascaris transfuga* egg (bear); (**e**) opisthorchiid egg (tiger); (**f**) *Capillaria* sp. egg (turkey); (**g**) hookworm egg (African civet); (**h**) *Toxocara* sp. egg (lion); (**i**) *Acuaria* sp. egg (peacock); (**j**) nematode larva (emu); (**k**) coccidian oocyst (emu); (**l**) *Eimeria* spp. oocyst (peacock).

**Figure 2 pathogens-10-00250-f002:**
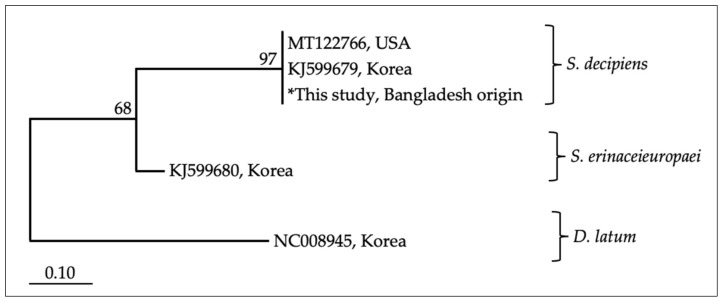
Phylogenetic relationships of *Spirometra* species based on *cox*1 gene sequences reconstructed using maximum likelihood (ML) method. Bootstrap values are shown above branches. The scale bar represents 0.10% divergence. (* indicates specimen used this study).

**Table 1 pathogens-10-00250-t001:** Foods and antiparasitic medications given to examined animals.

Animal	Scientific Name	Feed Given	Anthelmintic	Frequency
Horse	*Equus caballus*	Grass, cereal grains	Albendazole Or Triclabendazole with Levamisole	Every 3–4 months
Donkey	*Equus asinus*	Grass, cereal grains		
Black bear	*Ursus americanus*	Bread, mixed boiled feed (rice, milk, egg, fruits, vegetables)		
Spotted deer	*Axis axis*	Grass, cereal grains, vegetables		
Porcupine	*Hystrix indica*	Vegetables, grass, cereal grains		
Lion	*Panthera leo*	Meat	Ivermectin Or Albendazole	Every 3–4 months
African civet	*Civettictis civetta*	Meat, fruits		
Wild cat (bon biral)	*Felis silvestris*	Meat		
Royal Bengal tiger	*Panthera tigris*	Meat		
Hanuman langur	*Semnopithecus entellus*	Fruits, bread, vegetables, cereal grains,		
Monkey	*Rhesus macaque*	Fruits, bread, cereal grains, eggs		
Turkey	*Meleagris gallopavo*	Nuts, commercial poultry feed	Piperazine	Every 3–4 months
Peacock	*Pavo cristatus*	Nuts, vegetables, commercial poultry feed		
Owl	*Bubo bengalensis*	Meat, nuts		
Vulture	*Aegypius monachus*	Meat, nuts		
Ostrich	*Struthio camelus*	Fruits, bread, commercial poultry feed		
Emu	*Dromaius novaehollandiae*	Fruits, bread, commercial poultry feed		
Python	*Morelia spilota variegata*	Meat	Albendazole	Every 3–4 months

**Table 2 pathogens-10-00250-t002:** Eggs and oocysts found during coprological examination according to taxonomic order of hosts.

Type of Host	Host	Scientific Name	No. of Examined Samples	No. of Positives		Egg or Oocyst Observed	Size (μm) (*n* = 7)
				Helminths	Protozoa		
Mammals (Carnivores/Omnivores)	Lion	*Panthera leo*	2	2	-	*Spirometra* sp.	61–65 × 31–33
						*Toxocara* sp.	65–68 × 63–64
	Tiger	*Panthera tigris*	2	1	-	Opisthorchiid	26–29 × 13–14
	African civet	*Civettictis civetta*	3	3	-	*Spirometra* sp.	60–63 × 29–31
						Hookworms	68–72 × 39–41
	Bon biral	*Felis silvestris*	1	-	-	-	-
	Black bear	*Ursus americanus*	2	2	-	Hookworms	72–77 × 38–40
						*Baylisascaris transfuga*	63–65 × 58–60
	5 host species		10	8 (80%)	-	Genera of helminth: 5; Genera of protozoa: 0	
Mammals (Herbivores)	Donkey	*Equus asinus*	5	2	-	Gastrointestinal Strongyles	67–69 × 47–48
	Horse	*Equus caballus*	4	-	-	-	-
	Spotted deer	*Axis axis*	2	-	1	*Eimeria* spp.	35–36 × 19–24
	Porcupine	*Hystrix indica*	2	-	-	-	-
	Monkey	*Rhesus macaque*	4	3	-	Gastrointestinal Strongyles	74–78 × 43–46
	Hanuman	*Semnopithecus entellus*	2	2	-	Gastrointestinal Strongyles	68–71 × 45–48
	6 host species		19	7 (36.9%)	1 (5.3%)	Genera of helminth: 1; Genera of protozoa: 1	
Avian	Turkey	*Meleagris gallopavo*	2	1	1	*Capillaria* sp.	48–51 × 27–29
						*Eimeria* spp.	31–35 × 22–26
	Owl	*Bubo bengalensis*	2	-	-	-	-
	Vulture	*Aegypius monachus*	1	1	-	Roundworms	49–52 × 28–29
	Ostrich	*Struthio camelus*	4	-	-	-	-
	Peacock	*Pavo cristatus*	4	3	1	*Capillaria* sp.	49–51 × 28–29
						Roundworms	73–75 × 49–51
						Acuariid	34–36 × 17–19
						*Eimeria* spp.	29–31 × 18–21
	Emu	*Dromaius novaehollandiae*	2	1	1	Unidentified nematodes	length.: 2.1–2.4 mm width: 0.79–0.84 μm
						Coccidian oocysts	27–32 × 21–24
	6 host species		15	6 (40%)	3 (20%)	Genera of helminth: 4; Genera of protozoa: 2	
Reptile	Python	*Morelia spilota variegata*	1	-	-	-	-
Total	18		45	21 (46.5%)	4 (8.9%)		

“-“ indicates negative.

## Data Availability

All samples and genomic DNA were stored in the International Parasite Resource Bank (iPRB), Korea, and are available from the corresponding authors or first author (tilak.parasitology@sau.ac.bd) on reasonable request. The data are not publicly available due to privacy or ethical reasons.

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
