# Peer review of "Insight into One Health Approach: Endoparasite Infections in Captive Wildlife in Bangladesh"

_pathogens, 2021, doi:10.3390/pathogens10020250_

Round 1

Reviewer 1 Report

Dear Authors,

after a careful revision of your manuscript entitled “An Insight to One Health Approach: Intestinal Parasitic Infections from Captive Wildlife in Bangladesh”.

Despite the topic is of sound scientific interest in terms of “One Health” and the focus on critical points in the management and conservation of captive animals, along with the necessity to guarantee their well-being is attractive and important.

There are many major (and minor) issues that the authors should overcome.

Major revisions

It is this reviewer opinion that the authors cannot state with certainty the identity of the parasites detected through copromicroscopical examinations.

In particular, the authors performed molecular analysis to confirm the identity of the eggs only in the case of Spirometra spp. and it is not clear the reason of this choice. Even if the authors declare the inability to perform DNA extraction due to the deficiency of samples, this is an unsatisfying explanation, since based on my experience DNA can be extracted from small amounts of faeces.

The authors should provide satisfying reasons for the choice of performing or not the molecular analyses. In fact, molecular analyses are essential to confirm the identity of the parasite elements retrieved, especially in the cases where the morphological and morphometrical analyses did not allowed to recognize the species (e.g. “Hookworm”, Gatrointestinal strongyles, Opisthorchiid eggs, Capillaria spp. eggs). For instance, there are Capillaria and Hookworms species that are zoonotic and some other that are not zoonotic

Moreover, morphological and morphometrical keys used to identify the parasites are provided only for some of them, thus they must be integrated.

Since a molecular confirmation is lacking, it is impossible to draw specific conclusions, except for Spirometra decipiens.

Therefore, this reviewer strongly suggests to the authors to perform molecular analysis in order to confirm or not the identity of the parasites the captive animals sampled.

The title of the ms is misleading, as not only intestinal parasites have been found in the study (i.e. Opisthorchiid parasites). Moreover, it is better to focus the title on the importance of the management of parasitoses in captive animals through fecal examination and appropriate environment and dietary management, in the perspective of animal well-being and human health.

It is not totally correct speaking about “intestinal parasites”, as some of the adult forms of parasites retrieved do not live in the gastrointestinal tract (e.g. Opisthorchiid parasites).

The abstract is too long and contains an overload of unnecessary information, while for instance important data on animal sampled are not provided, thus the authors should reduce the abstract and specify which animals are included in the “18 different species of wild animals” (i.e. 11 species of mammals: herbivores, carnivores and omnivores, one species of reptile and 6 species of birds).

The authors provide information on the pathogenic power in humans and animals only for some parasites, thus they should be integrated to define a comprehensive framework to the reader.

It is in the opinion of the reviewer that data collected from questionnaires and interviews to veterinarians and zoo workers are collateral and out of the scope of the study. Moreover, the sample size is too small to draw any conclusion. Thus, these data should be completely deleted from the present study and used with other purposes. Alternatively, the authros could increase the sample size.

Besides the appropriate considerations on the anthelmintic resistance due to the routinely treatments and the indications on the correct environmental and dietary management of the animals, the concept of routinely fecal examinations with the consequent appropriate treatment should be emphasized.

The authors should also cross-reference the positivity to parasites and the treatment provided to the animals, discussing possible reasons of lack of efficacy, other than drug-resistance (e.g. drug not effective or not registered for certain parasites).

Discussions should be completely rewritten on the basis of the molecular analyses, if they could be performed by authors.

Minor revisions

There are some minor revisions that should me made throughout the ms:

Please write the genera name of parasites in cursive/italics, e.g. in line 205, 212 and 213.

Some redundant sentences are present and should be avoided.

L 215-216: The sentence on the parasite found in emu faeces is not sufficient, the authors should give some hypothesis on the identity of this larva (e.g. it could be a pathogenic species? Which one? It could be a free-living roundworm larva?)

L 234: The authors should delete the quotation marks at the beginning of the sentence.

L 244: add another point after “sp.”.

The font size should be the same in all the ms. Please modify the size where necessary, e.g. lines 338, 343, 389, 400, 415.

Reviewer 2 Report

Authors have made changes according to the previous review. However in my opinion the manuscript still needs some corrections.

l.119-120 – carnivores instead of carnivore

  1. 117-124 – the prevalence of hookworms should also be mentioned in the paragraph

l.124 – I would add “unidentified coccidian oocyst”

l. 148 – I am aware….

l. 188-190 – this should be clearly stated that raw meat or fish, which in many cases may be the direct source of infection as these may be tissues of intermediate or paratenic hosts and contain larval forms of parasites.

l. 196 - According to one health concept another aspect should be mentioned here, that is the protection of the environment form contamination with parasite eggs from improperly disposed animal feces. This is one of the key steps to stop the transmission of parasites to the wildlife.

l.199 – rodents may also be paratenic hosts for other parasitic nematodes identified in this study, such as Toxocara sp, and hookworms so these also should be mentioned.

l. 215-216 – I would rewrite this sentence – We were not able to identify the species of nematode larvae found in emu feces which measured….Please remove quotation marks.

l. 226-229 – tissues of intermediate hosts could be the direct source of infection in these cases. If animals were fed with raw fish these could have been infected with cestode or trematode larvae. This possibility should also be mentioned.

l. 234 – remove quotation mark

l. 255 – only Toxocara cati can be found in a lion, please remove T. canis

l. 267-268 – please specify – eating raw fish was the source of infection but inadequate sanitation and improper disposal of feces leads to the continuation of the life cycle by contamination of the environment and infection of intermediate hosts.

l. 276-277 – animals mentioned here may serve as paratenic hosts for most zoonotic parasite species identified in the study. Stray dogs and cats will serve mainly as a source of roundworm and hookworm eggs which may infect animals living in the zoo. I would say that roundworms and hookworms are more prevalent and pathogenic to carnivores, than Siprometra tapeworms and deserve some more attention from the animal health point of view. Toxocara was only mentioned from the human health perspective and hookworms were not mentioned at all. Moreover hookworms are also zoonotic nematodes. Hookworm species infecting animals may also infect humans by direct contact (skin penetration) causing cutaneous larva migrans. The zoo staff is at risk of hookworm infection during contact with animals or during cleaning of animal enclosures. I suggest adding more information on Toxocara roundworms and hookworms with appropriate references.

Round 2

Reviewer 1 Report

Dear authors, 

unfortunately, mere answers to this reviewers revision are not satisfying. More importantly, I can see no improvements in your manuscript. That is still not suitable for publication on Pathogens to my opinion.

Kind regards
